# Heat Shock Proteins in Tooth Development and Injury Repair

**DOI:** 10.3390/ijms24087455

**Published:** 2023-04-18

**Authors:** Shuling Guo, Haosun Yang, Jiacheng Liu, Zhaosong Meng, Lei Sui

**Affiliations:** School of Stomatology, Tianjin Medical University, Tianjin 300014, China; shulingguo@tmu.edu.cn (S.G.); 13212014386@163.com (H.Y.); liujiachengboy@126.com (J.L.)

**Keywords:** heat shock protein, tooth development, tooth injury repair, signaling pathway

## Abstract

Heat shock proteins (HSPs) are a class of molecular chaperones with expression increased in response to heat or other stresses. HSPs regulate cell homeostasis by modulating the folding and maturation of intracellular proteins. Tooth development is a complex process that involves many cell activities. During tooth preparation or trauma, teeth can be damaged. The damaged teeth start their repair process by remineralizing and regenerating tissue. During tooth development and injury repair, different HSPs have different expression patterns and play a special role in odontoblast differentiation and ameloblast secretion by mediating signaling pathways or participating in protein transport. This review explores the expression patterns and potential mechanisms of HSPs, particularly HSP25, HSP60 and HSP70, in tooth development and injury repair.

## 1. Introduction

Heat shock proteins (HSPs) are a class of molecular chaperone proteins that are widely expressed in various cells. On stimulation by high temperature, hypoxia, drugs, heavy metals, or other stimuli that affect cell homeostasis, HSPs bind to their associated proteins via structural changes, affecting the folding and maturation of intracellular proteins. Therefore, HSPs can prevent protein aggregation and degrade abnormal proteins, enhancing cell resistance to stimuli and regulating cell homeostasis [1]. Previously, the nomenclature of HSPs was divided based on its molecular weight into HSP110, HSP90, HSP70, HSP40 and sHSP; however, based on the nomenclature of the HUGO Gene Nomenclature Committee (HGNC), human HSPs were named HSPA (HSP70), HSPB (sHSP), HSPC (HSP90), HSPD (HSP60), HSPH (HSP110) and DNAJ (HSP40) [2]. Further classification of HSPA includes HSPA1A (HSP70-1), HSPA5 (GRP78, glucose-regulated protein 78) and HSPA8 (HSC70); HSPB includes HSPB1 (HSP27, murine homologue HSP25); HSPD includes HSPD1 (HSP60). Studies on mice showed HSP70 plays an important role in maintaining hematopoietic stem cells by controlling oxidative stress [3]. Moreover, overexpression of sHSP, such as HSP25 in Drosophila melanogaster, was found can lead to a mild increase in lifespan [4]. Additionally, the expression of HSP is very important for the normal progression of pregnancy [5]. In organs, such as the heart and liver, some HSP also plays a role in protecting tissues from damage. So, previous studies on different aspects have all revealed the function of HSP for maintaining cell homeostasis to various stimuli. On the other hand, studies have demonstrated the role of HSPs in the occurrence of diseases such as cancer, neurodegenerative diseases and atherosclerosis [6,7,8]. Numerous studies have reported on the roles of HSPs during tooth development and tooth injury repair, demonstrating that HSPs have different expression patterns at different stages of tooth development. Tooth development is a complex physiological process involving cell proliferation, differentiation, and the formation of enamel and dentin, among others, during which cellular activities are more vigorous. In addition, tooth development is a dynamic process; with the change in tooth germ morphology, the environment of the cells in the tooth germ also changes, leading to environmental stimulation for these cells. HSPs play an important role in tooth development by maintaining cell homeostasis in both of the above situations. Whereas when a tooth is injured, the tooth undergoes repair by remineralization and regeneration. Similar to the process of tooth development, damage causes sudden changes in the cellular environment and cellular activities are greatly intensified during repair; HSPs similarly exert their effects to maintain cell homeostasis during tooth injury repair. However, the difference is that, during tooth development, the stimulus to the cells mainly comes from the change of the internal microenvironment, while in injury repair, the external stimulus dominates and may evoke an immune response. Among the numerous HSPs related to tooth development and injury repair, there are many studies on HSP25, HSP60, and HSP70, which will be discussed in detail in the following text. This article aims to review the expression patterns and possible mechanisms of HSPs in the tooth, providing a foundation for elucidating the role of HSPs in the field of tooth development and injury repair.

## 2. HSP25

### 2.1. Expression of HSP25 in the Developing Molar

HSP27 (murine homolog HSP25) is ubiquitously expressed in all tissues of the human body. The human HSP27 gene sequence maps to 7q11.23 and comprises 205 amino acid residues. The expression of HSP25 is increased when cells are stimulated by hypoxia, heavy metals and mechanical stress [9,10,11], which maintain cell stability and improve cell resistance to internal and external stimuli by protecting the actin and cytoskeleton [11], inhibiting apoptosis, enduring hypoxia [9] and reducing protein misfolding and aggregation [12]. The roles of HSP25 and several other HSPs, when cells are stimulated by the external environment, are shown in Figure 1. HSP27 could alleviate heart aging through antioxidant and mitochondrial activation [9]. High HSP25 levels are able to reduce the occurrence of cardiovascular events. Additionally, overexpression of HSP25 can reduce cholesterol levels in blood and plaque, as well as the development of inflammation and atherosclerosis [13]. In nonalcoholic fatty liver disease, phosphorylated HSP25 is able to interact with STAT3 (Signal transducer and activator of transcription 3), thereby stimulating autophagy and lipid droplet clearance [14]. The expression of HSP25 varies with the stage of tooth development in the tooth germs and other surrounding tissues.

Murine molars are similar to permanent human teeth. Nakasone et al. [15,16] found no HSP25 expression during the bud-to-bell stage of the molar tooth germs, which is during the 15–18 embryonic days (E15–18); however, the oral epithelium, except the basal layer, showed HSP25 expression. At the late bell stage (E20), HSP25 is expressed in the stellate reticular layer with no expression in other components of the tooth germ. On the day of birth, HSP25 is expressed in the apical region of the dental papilla, weakly expressed in pre-odontoblasts, strongly expressed in odontoblasts and mesenchymal cells in the sublayer of odontoblasts with continued expression in the stellate reticular layer, but not in other parts of the tooth germ. On tracking HSP25 expression in molars from postnatal (PN) day 1 to 100 in rats, HSP25 was found to be mainly localized in the pulp horn, while pre-ameloblasts and pre-odontoblasts were weakly positive for HSP25, with HSP25 strongly positive at the distal end of pre-ameloblasts than the proximal end, and odontoblasts with mesenchymal cells were strongly positive for HSP25 on PN1. On PN5, which is the secretory stage of ameloblast, odontoblasts and ameloblasts abundantly expressed HSP25, while mesenchymal cells in the pulp horn region showed a weak expression. On PN10, the mature stage of ameloblast, HSP25 expression in ameloblasts further increased, while the expression in the remaining cells remained unchanged. On PN15, in the initial stages of root formation, HSP25-positive areas move from the pulp horn to the root, while odontoblasts and ameloblasts continue to express HSP25 strongly. On PN30, when the root apex begins to close, the odontoblasts close to the crown strongly express HSP25, with a gradual decrease towards the root. On PN60–100, when the root formation is complete, HSP25 is expressed in the odontoblasts. The expression pattern of HSP25 from PN1–100 in mouse molars is shown in Figure 2. On co-localization with the cell proliferation marker BrdU, which made the crown-to-root structure visible, BrdU-positive cells gradually increased as the tooth developed, which is contrary to the expression trend of HSP25. Since the cellular process of dentin or enamel secretion by odontoblasts and ameloblasts begins after they exit the cell cycle to complete differentiation, pre-odontoblasts and pre-ameloblasts show weak HSP25 expression after their proliferative activity stops, followed by strong HSP25 expression by odontoblasts and ameloblasts, indicating that HSP25 is involved in the differentiation process during tooth development. Therefore, HSP25 has the potential to be a switch marker between cell proliferation and terminally differentiated cells during tooth development. Odontoblasts and ameloblasts display changes in cell morphology and organelle distribution during differentiation, and their mechanism of action may be related to the changes in the cytoskeleton [17,18]. HSP25 regulates the cytoskeleton and actin filaments during differentiation, maintains cell stability during protein synthesis and secretion after cell maturation, and aids odontoblasts and ameloblasts’ functionality.

Onishi et al. [19] conducted a study on HSP25 expression during root development in rat maxillary first molars, which showed that HSP25 is expressed in ameloblasts and endothelial cells along the root surface, fibroblasts in the odontoblast’s sublayer and odontoblasts in the pulp of the periodontal ligament of rats on PN14. On PN21, when the first molar erupts, HSP25-positive cells appear intermittently close to the primary cementum and odontoblasts, while the sub dentin layer cells and vascular endothelial cells continue to express HSP25. On PN28, when the roots of the first molar are almost fully developed, HSP25 shows continued expression by cells close to the primary cementum and odontoblasts; however, HSP25-positive cells were not observed along with the secondary cementum. Epithelial rests of Malassez (ERM) cells express HSP25 in the cervical and bifurcation regions of the tooth. Studies have shown that ERM cells are more active in the bifurcation region than the cervical region, and some may be embedded in the cementum to become cementocytes [20]. Therefore, HSP25 has the potential to regulate ERM cells’ activity.

Additionally, a study by Du et al. [21] on HSP25 expression in the dental follicle of rat mandibular first molars showed that HSP25 expression was upregulated during dental follicle development. Moreover, on PN5, HSP25 was slightly expressed in the dental follicle of mandibular first molars. On PN5–7, HSP25 expression remained weak. However, HSP25 expression increased from PN7 onwards and was strongly expressed by PN9–11. In vitro cultures of dental follicle cells revealed that HSP25 was expressed only in the cytoplasm, with HSP25 showing no significant effect on cell proliferation; however, it has been speculated to play a role in cell differentiation. Additionally, on culturing murine dental follicle cells in vitro with interleukin-1 α (IL-1 α) and colony-stimulating factor-1 (CSF-1), a significant up-regulation of the cytoskeleton and HSP25 expression was detected [22], indicating that HSP25 expression in the dental follicles could be associated with the development-related morphological changes (Table 1 and Table 2).

### 2.2. The Expression of HSP25 in the Developing Incisor

Murine incisors continue to develop after birth; therefore, they are considered an ideal model for studying tooth development. Nakasone et al. [16] found that HSP25 is first expressed at the differentiation stage in the incisors, with localization mainly in the intermediate layer, mesenchymal cells, pre-odontoblasts, odontoblasts, pre-ameloblasts and ameloblasts. The roots of rodent incisors have been reported to contain mesenchymal stem cells [23]; however, Nakasone et al., did not observe HSP25 expression in this region, suggesting that adult stem cells in murine incisor roots may not exhibit HSP25 immunoreactivity. Additionally, HSP25 expression was not expressed in enamel knots, indicating that enamel knot cells may not complete terminal differentiation and are prone to apoptosis, which reflects their disappearance in the later development stage. Ohshima et al. [24] report on HSP25 expression in the incisors of PN30 rats. In the early stage of ameloblast secretion, pulp cells present in the pulp horn and odontoblasts express HSP25, with intense, localized HSP25 expression only in the odontoblasts cytoplasm and no expression in developing organelles, such as Golgi apparatus, rough endoplasmic reticulum (ER) and mitochondria. Ameloblasts weakly express HSP25 in the mesial and distal ends of the cell, while stratum intermedium cells show expression in the cytosol; however, HSP25-positive cells were not observed in the periodontal ligament. Subsequently, during the secretory stage of ameloblasts, HSP25 was strongly expressed in ameloblasts, while stratum intermedium cells became HSP25-negative, and the other cells’ HSP25 expression remained unchanged. At the mature stage of ameloblasts, strong HSP25 positivity was seen in ameloblasts and odontoblasts, along with some blood vessels in the periodontal ligament and mesenchymal cells in the odontoblast sublayer. HSP25 expression trends in the incisors of PN30 rats are shown in Figure 3. During dentin secretion, odontoblasts show high dentin secretion and have high RNA and protein concentrations in the cytoplasm, wherein HSP25 acts as a chaperone and prevents protein aggregation during and after protein synthesis [25]. Odontoblasts retreat inward as dentin deposition increases, owing to the mechanical stress from the increased cell density. This microenvironmental change induces strong HSP25 expression in these cells. In ameloblasts, HSP25 expression is increased as the secretory activity of the cells begins. HSP25 reactivity is stronger in ruffle-ended ameloblasts with exuberant secretion than in smooth-ended ameloblasts, indicating that the formation and maintenance of ameloblast boundaries at the ruffled ends could be related to HSP25. Additionally, co-localization of HSP25 with actin filaments was observed in odontoblasts and ameloblasts. During dentin and enamel secretion, both odontoblasts and ameloblasts move back due to the accumulation of dentin and enamel. It has been demonstrated that HSP25 stabilizes actin filaments and cytoskeleton during ischemic acute renal failure [26] and oxidative stress in the intestinal mucosa [27]. Additionally, Ohshima et al. [24] demonstrated that HSP25 was co-localized with actin filaments in odontoblasts, suggesting that HSP25 plays a role in maintaining the integrity of odontoblasts during mechanical stress by regulating actin dynamics or apoptosis. In secretory ameloblasts, HSP25 is mainly localized in the terminal web, which contains actin, myosin and tropomyosin. During enamel secretion, ameloblasts gradually move back with the terminal network acting as a power source for their motility [28]. Therefore, HSP25 expression, observed in ameloblast cytoplasm, plays a role in the formation and maintenance of actin filaments by regulating actin, thereby maintaining the integrity of the cell layer during enamel secretion (Table 3).

### 2.3. Expression of HSP25 in the Gingiva during Tooth Eruption

Sasaki et al. [29] studied HSP25 expression in the gingiva during tooth eruption, reporting a consistent expression before and after tooth eruption. In the junctional and oral mucosal epithelium, the spinous and granular layers of the stratified scale epithelium express HSP25. Additionally, the stratified scale epithelium partially facing the crown enamel and the simple cuboidal epithelium formed by the enamel epithelium shows HSP25 immunoreactivity. At the time of tooth eruption, HSP25 is mainly localized in the spinous and granular layers of the gingival epithelium. Expression of HSP25 in the gingiva was also shown in Figure 2. Using Ki67 to label proliferating cells, no co-localization of HSP25 with Ki67 was observed in the gingiva at or after tooth eruption. In the stratified scaly epithelium, cell proliferation occurs mainly in the basal layer and reaches the epithelial surface layer superiorly through the spinous and granular layers, differentiating into keratinocytes. Studies on skin demonstrated that HSP25 expression gradually increased as keratinocytes moved away from the basal layer to the epidermal layer [30], indicating that HSP25 is involved in the keratinocyte differentiation process [31]. HSP25 was found to be highly expressed in resting keratinocytes [32], which is consistent with the fact that HSP25-expressing cells in the gingival epithelium are confined to the spinous and granular layers. Therefore, it can be speculated that the expression pattern of HSP25 in the gingival epithelium is related to the differentiation of keratinocytes and inhibition of their apoptosis. Pemphigus vulgaris is a disease characterized by keratinocyte separation, and it has been demonstrated that HSP25 is involved in the p38 mitogen-activated protein kinase (p38-MAPK) function and inhibits keratinocyte separation, thereby preventing pemphigus vulgaris [33]. Therefore, HSP25 expression in the gingiva could be related to the p38-MAPK mechanism of action.

### 2.4. Expression of HSP25 during Tooth Injury

Tooth damage occurs during cavity preparation [34] and tooth replantation. HSP25 expression in the pulp and the relationship between HSP25-positive cells and major histocompatibility complex class II (MHC-II) positive cells after cavity preparation were studied by Ohshima et al. [35] and Suzuki et al. [36]. They found that the odontoblasts in the injured area were severely damaged after cavity preparation, with HSP25 expression gradually decreasing until it completely disappeared as the odontoblasts degenerated. At 12 h after surgery, class II MHC-positive cells were arranged along the pulp-dentin margin, with their processes extending deep into the dentinal tubules, which expressed HSP25. HSP25 expression was found in some mesenchymal cells at 24 h after cavity preparation, while class II MHC-positive cells showed no immunoreactivity with HSP25 at the pulp-dentin margin. At 72 h after cavity preparation, newly differentiated odontoblasts strongly expressed HSP25 and aligned in the odontoblast layer. Furthermore, Kawagishi et al. [37] reported that the formation of third-stage dentin by next-generation odontoblast-like cells could be detected by observing the degeneration and regeneration of HSP25-positive cells after tooth replantation. These studies further confirm that HSP25 can act as a switch between cell proliferation and cell differentiation during odontogenesis and amelogenesis. Moreover, HSP25 in the disrupted odontoblasts could aid in the accumulation of class II MHC-positive cells at the site of injury.

### 2.5. HSP25-Related Signaling Pathways during Tooth Development

Tooth development involves various signaling pathways, such as Wingless/Int (WNT), Bone Morphogenetic Protein (BMP), Fibroblast Growth Factor (FGF) and Sonic hedgehog (Shh) signaling pathway [38]. Among them, extracellular regulated-signal kinase (ERK), a subfamily in the MAPK pathway [39], is involved in FGF signaling [40], and Lee et al. [41] found that HSP25 regulates mouse incisor development by activating the ERK/MAPK signaling pathway and transducing this signal to the ameloblasts, while phosphorylated ERK acts as a positive regulator of HSP25 expression. HSP25 controls the activity of ameloblasts derived from the inner enamel epithelium and is down-regulated in the MAPK signaling pathway during ameloblast terminal differentiation. Additionally, p38 MAPK, another subfamily of the MAPK signaling pathways, has been reported to play a crucial role in tooth development [42,43]. Studies have shown that HSP25 acts as a downstream target in the p38 MAPK pathway in response to mechanical stress. The mechanism is that HSP25 is phosphorylated by mitogen-activated protein kinase-activated protein kinase 2 (MAPKAPK2), and the phosphorylated HSP25 can attach to the actin cytoskeleton, thus maintaining cell stability under mechanical stress [44]. Referring to the mechanical stress-induced HSP25 expression in odontoblasts and ameloblasts during tooth development, HSP25 is speculated to play a role in tooth development in which p38 MAPK is involved. Furthermore, Eapen et al. [45] found that dentin matrix acidic phosphoprotein 1 (DMP1) could activate and up-regulate the expression of HSP25 in pre-osteoblasts via the p38 MAPK pathway during tooth development. HSP25 also induces focal adhesion kinase and ERK phosphorylation through integrin β1-mediated signaling, thereby increasing cell adhesion and migration in the odontoblastic cell line MDPC-23 [45]. HSP25-related signaling pathways in tooth development and injury repair are shown in Figure 4.

## 3. HSP60

### 3.1. Expression of HSP60 in the Developing Tooth

HSP60 is a class of chaperone proteins mainly present in cellular mitochondria, and the human HSP60 gene sequence maps to 2q33.1. HSP60 plays an important role in resisting apoptosis [46] and assisting protein folding [7]. Additionally, it plays a role in immune regulation and can be induced to activate immune cells [8]. Therefore, current research on HSP60 has focused on its activation of immune cells and maintenance of mitochondrial function. Studies have shown that HSP60 can induce dendritic cells, which contributes to the formation of an atherosclerotic plaque [8]. HSP60 acts as a chaperone in mitochondria; it has been shown that in renal tubular dysfunction due to diabetes, HSP60 plays a role in regulating intracellular protein aggregation, ATP production and oxidative stress in renal tubular cells [47]. In addition, HSP60 has an important role in maintaining cardiac physiological function. HSP60 loss may lead to ventricular dilatation and heart failure. At the same time, HSP60 can be used as a biomarker of heart failure [48]. On the other hand, the effect of HSP60 on cellular energy supply also plays a role during the growth of tumors [49], and its mutation may result in impaired cellular respiration and insufficient ATP production, which can lead to neuropathy and paraplegia [7]. Moreover, HSP60 has different expression patterns during tooth development, affecting the morphology of tooth germs. However, studies regarding the role of HSP60 in tooth development are scarce, and there’s no study on the expression of HSP60 in tooth injury.

The expression of HSP60 in embryonic mouse incisors and its effect on tooth morphology was studied by Papp et al. [50]. HSP60 first appears in the epithelial belt during the initial stages of tooth development at E11.5. From the bud stage to the cap stage (E13.5–15.5), HSP60 is expressed in the inner enamel epithelium, outer enamel epithelium and enamel knots, with low HSP60 expression in the dental papilla and dental follicle. During the bell stage (E16.5–18.5), HSP60 is expressed in the inner enamel epithelium, outer enamel epithelium, pre-ameloblasts, ameloblasts and intermediate layer cells and begins to increase in pre-odontoblasts and odontoblasts, from E16.5. Generally, HSP60 mRNA is distributed mainly on the labial side of the tooth germs, with less distribution on the lingual side, and a small amount of HSP60 mRNA can be detected in both the dental follicle and surrounding tissues. Treatment of in vitro cultured tooth germs with exogenous excess HSP60 revealed that the distance between the labial and lingual cervical rings of the tooth germs was closer than that of the control tooth germs, while the included angle between the enamel knot and the two cervical rings was blunt. Pandya et al. [51] found that HSP60 expression in mouse molar ameloblasts peaked on PN2 and gradually decreased on PN4 and PN6. Expression patterns of HSP60 in mouse molar from PN1–4 were shown in Figure 2. Moreover, it is speculated that HSP60 is associated with preventing amelogenin misfolding (Table 4).

### 3.2. HSP60-Associated Signaling Pathways during Tooth Development

Osteoclasts play an important role in tooth development [52]. IL1β and tumor necrosis factor (TNF)-α has been reported to stimulate osteoclasts for HSP60 secretion and production, with HSP60 enhancing osteoclast production and bone resorption [53]. HSP60 has application potential that can be used to affect tooth development morphology. Additionally, it has been found that in maternal diabetes, the Toll-like receptor 4 (TLR4)/nuclear factor κB (NF-κB) pathway is activated and affects the proliferation and apoptosis of embryonic odontogenic epithelium and intercellular cells [54]. HSP60 affected apoptosis by activating the NF-κB pathway mediated through TLR4 [55]. This leads to the speculation that HSP60 participates in the TLR4/NF-κB pathway to affect tooth development. HSP60 can also bind to the IKK (IκB kinase) complex [56], which contains IKKα, IKKβ and IKKγ. Studies have shown that IKKα is involved in tooth development by influencing epithelial invagination through participating in the Notch signaling pathway during the early stage of tooth development. Additionally, the absence of IKKα can cause abnormal tooth developmental morphology. HSP60 may therefore affect tooth developmental morphology through IKKα in the early stage of tooth development [57].

## 4. HSP70

### 4.1. Expression of HSP70 in the Developing Tooth

HSP70 is a class of chaperone proteins that contains the most members, and according to classification rules [2], the HSP70 family mainly contains HSP70-1, GRP78 (glucose-regulated protein 78) and HSC70. The HSP70 gene sequence maps to 6p21.33, and its roles include delaying protein folding to prevent protein misfolding, aiding transmembrane transport, and inhibiting aggregation and dissociation of aggregates [58]. Thus, in acute infections, such as multiple organ dysfunction syndrome and disseminated intravascular coagulation caused by sepsis, HSP70 functions to repair misfolded proteins, thus resisting oxidation and attenuating inflammatory responses [59]. It also forms an HSP70 chaperone network during protein folding by cooperating with other chaperones, such as HSP60 and HSP90 [60]. In hepatic ischemia/reperfusion (I/R) injury, HSP70 is upregulated by inducible nitric oxide synthase (iNOS) to protect against I/R injury [61]. In renal, changes in the concentration of sodium and urea in the renal medulla can induce HSP70. Thus, HSP70 is associated with maintaining osmotic pressure and may be implicated in the development of essential hypertension [62]. Further, HSP70 is associated with the development of cancer and certain neurodegenerative diseases [63].

GRP78 is an ER chaperone whose gene sequence maps to 9q33.3. ER stress occurs when misfolded and unfolded proteins accumulate in the ER; however, ER burden is reduced on GRP78 activation, which degrades misfolded proteins and increases ER content. In cardiomyocytes, GRP78, together with its partner BAG5 (Bcl-2 associated athanogene 5), alleviates cardiomyocytes apoptosis caused by ER stress [64]. Additionally, GRP78 could protect liver hepatocytes from ER stress by interacting with GALNT6 (Polypeptide N-Acetylgalactosaminyltransferase 6) [65]. GRP78 also plays a role in many pathological processes. GRP78 is associated with the proliferation, invasion and metastasis of cancer cells, such as in breast and colon cancer. GRP78 is also present on the cell surface and acts as a receptor to interact with a variety of ligands and proteins. Studies have shown that cell surface GRP78 can elicit autoimmune responses and further confirmed that cell surface GRP78 is associated with atherosclerosis and rheumatoid arthritis [6].

HSC70 is a constitutively expressed chaperone protein whose gene sequence maps to 11q24.1. It plays a role in the prevention of protein misfolding, presentation of antigenic peptides by class II MHC-positive cells to CD4+ T cells and cellular autophagy [66]. Another role of HSC70 is to participate in intracellular protein trafficking, and chaperones import proteins from the cytoplasm into the nucleus or organelles in an ATP-dependent manner. On the other hand, HSC70 has been implicated in cellular protein degradation and autophagic processes [67]. Studies have shown that in acute kidney injury, HSC70 interacts with legumain to promote autophagy, thereby promoting apoptosis of harmful cells and reducing kidney injury [68]. Various studies on the above three HSP70 members have been reported; however, this review focuses on the expression and role of HSP70, GRP78 and HSC70 in tooth development and injury repair.

A study by Kero et al. [69] regarding HSP70 expression in the human incisor tooth germs showed that HSP70 was strongly expressed at embryonic week 9 in the incisor tooth germs, dental lamina, inner enamel epithelium and outer enamel epithelium but weakly expressed in the primary enamel knots. HSP70 expression in the tooth germs at embryonic week 12 was similar to that at week 9, with moderate expression in the cervical ring and stellate reticular layer. At embryonic week 14, HSP70 expression was attenuated in the dental lamina, inner enamel epithelium and stellate reticular layer in the incisor tooth germ, while expression in the outer enamel epithelium and cervical ring remained unchanged. Additionally, HSP70 was weakly expressed in the dental papilla and intermediate layer and moderately expressed in pre-ameloblasts. At week 20, HSP70 expression in the incisor tooth germs was confined to the enamel organs. HSP70 expression in the dental lamina was further attenuated, with the stellate reticular layer, intermediate layer and inner enamel epithelium showing moderate expression, while the outer enamel epithelium showed weak expression. The cervical ring, pre-ameloblasts and the scattered odontoblasts in the dental papilla showed strong HSP70 expression. Thus, the enamel organ of the human incisor germs shows high HSP70 expression, which is associated with the high proliferative activity in this region, possibly due to the continuous mechanical stress caused by the growth of cells towards the mesenchymal tissues. Furthermore, the disintegration of the primary enamel knot correlated with the reduction of HSP70 expression, which can be attributed to the inhibition of apoptosis by HSP70. A study by Ravindran et al. [70] showed that during the embryonic period, GRP78 is mainly localized in the pre-odontoblasts and pre-ameloblasts and abundantly expressed in the undifferentiated cells of the cervical ring. After birth, GRP78 was mainly localized in the cytoplasm of osteoblasts, odontoblasts and ameloblasts in the developing incisors in mice on PN1 and PN3. In developing molars, GRP78 was mainly expressed in pre-odontoblasts and pre-ameloblasts. Since GRP78 is a secreted protein, it is also localized in the dentin and alveolar bone. On PN5, GRP78 localization in the anterior part of the mineralization of the dentinal matrix in the incisors and alveolar bone matrix was observed. On PN7, the entire pulp, dentin, bone matrix and ameloblasts showed GRP78 localization. On PN20, GRP78 was mainly localized in the pre-dentin and periodontal ligament, with no GRP78 expression in the odontoblast. Therefore, the expression and secretion of GRP78 increase only when the matrix is secreted and mineralized. The expression trend of GRP78 in the molar ameloblasts from PN2–6 in mice was consistent with that of HSP60, with both proteins peaking on day two and then gradually decreasing. However, HSP70 expression rapidly decreased after peaking on PN2 and was absent by PN4. Expression patterns of HSP70 and GRP78 in mouse molar from PN1–4 were shown in Figure 2. This is consistent with Sens et al. [71] reported the absence of HSP70 expression in the pulp of mature human molars; however, HSC70 was found to be highly expressed in the odontoblasts and fibroblasts, moderately expressed in the endothelial and smooth muscle cells with localization in the nucleus and cytoplasm and weakly expressed in the Schwann cells with localization only in the nucleus (Table 5).

### 4.2. Expression of HSP70 in Tooth Injury

HSP70 expression increases responsively within 24 h of tooth injury [72]. Austin et al. [73] studied macaques and found that when macaques are subjected to external stimuli or social stress, bands are formed with the deposition of elements, such as barium, strontium, lead and zinc, in the dentin. Moreover, the increase in HSP70 expression causes a co-localization with these element signals after their release, indicating that this increase in HSP70 expression is a response to the endogenous metals released during bone remodeling. Further, when hydroxyethyl methacrylate (HEMA) is used to repair teeth, it can lead to blocked mineralization, causing local inflammation and inducing an ER stress response owing to increased GRP78 expression [74]. Additionally, the immune response can maintain high expression of HSPA9 (HSP70 family member), thereby enhancing the osteogenic potential of dental pulp stem cells and gingival stem cells by activating the cytoskeletal remodeling [75].

### 4.3. HSP70-Related Signaling Pathways during Tooth Development

GRP78 has been demonstrated to mediate endocytosis of DMP1 [76]. Furthermore, the tyrosine phosphorylation of GRP78 is a critical condition in DMP1- and GRP78-mediated calcium release [73]. DMP1 plays an important role in the maturation and secretion of odontoblasts and osteoblasts. DMP1, Krüppel-like factor 4 [77] and fibroblast growth factor (FGF) [78] affect the differentiation of odontoblasts. DMP1 and FGF are also involved in the signaling pathways within MAPKs, such as ERK, MEK and BMP signaling pathways [78]. The role of GRP78 in this process is unclear; however, GRP78 is speculated to play a role in these pathways. HSP70 has been pointed out to promote osteogenesis of human stem cells by activating the ERK signaling pathway [79]. Moreover, overexpression of HSP70 enhances the osteogenic differentiation of bone marrow mesenchymal stem cells via activation of the Wnt/β-catenin signaling pathway and increases the expression of osteogenesis-related genes, such as runt-related transcription factor 2 (Runx2) [80]. Considering all the ERK and Wnt/β-catenin signaling pathways are involved in tooth development, it can be speculated that Hsp70 plays a role in tooth development through the above signaling pathways. Studies on the role of the other members of the HSP70 family in signal transmission during tooth development are scarce.

## 5. Research Prospect of HSPs in Tooth Development and Injury Repair

The current research on HSPs in terms of tooth development and damage repair is inadequate. More studies on how HSPs are expressed are needed. Currently, the role of HSPs in these processes is unclear, with various speculations regarding the many possible mechanisms of action based on the role of HSPs in other tissue cells. The signaling pathways that HSPs may participate in tooth development and injury repair are shown in Figure 5. HSPs act as molecular chaperones and play a role in many cellular processes, but these roles are mainly auxiliary. This has led to HSPs being considered complementary research subjects in various studies. HSPs are also often used as markers to verify the emergence of certain phenomena. Additionally, the role of HSPs has been demonstrated in processes such as the tumorigenesis [6,49,63], alleviating tissue damage under-stimulation [26,27] and stem cell protection [81], providing a practical value for the study of HSPs. Regarding the practical applications, subsequent studies on tooth development and damage repair can be used to (1) investigate more the common roles of HSPs during tooth development; (2) validate the HSP-involved signaling pathways during tooth development and injury; (3) discover unique signaling pathway processes in the oral cavity; (4) explore various types of HSPs that can be used as important markers during the development of other teeth; (5) discover the relationship between HSPs and dental developmental diseases; (6) investigate the role of HSPs in tooth restoration and tooth damage reduction in clinical treatment; (7) explore the role between HSPs and odontogenic stem cells. However, there are still some issues that have not been mentioned in this review. As a class of chaperone proteins comprising many members, HSPs have a wide range of roles and many mechanisms of involvement, making it difficult to explain their complete, complex roles. In addition to various speculations and hypotheses based on existing theories, further research is required to obtain a more in-depth understanding of the basic structure and basic mechanism of action of various types of HSPs.

## 6. Possible Roles of HSP10, HSPB5, HSPB6, HSPB8, HSP90 and HSP110 in Tooth Development

As mentioned earlier, the basic role of various types of HSPs is to maintain cellular homeostasis and improve the resistance of cells in the face of different stimuli. Due to different expression positions and molecular structures, HSPs achieve the purpose of maintaining cell homeostasis through different pathways. However, some HSPs, such as HSP60 and GRP78, also play a role in the development of the disease. The role of HSPs for tissues, therefore, needs to be considered from both physiological and pathological aspects. Since the study of HSPs is very limited in the field of tooth development and injury repair. Only HSP25, HSP60 and HSP70 have been studied more, so this review only covers HSP25, HSP60 and HSP70. Additionally, previous research mainly described the expression phenomenon of HSPs during tooth development and injury repair. The understanding of the mechanism of HSPs in tooth development and injury repair is very limited and can only be speculated by the role of HSPs in other tissues. There are still many unmentioned HSPs that also play important functions in life activities.

For the sHSP family, the combination of HSP10 and HSP60 is able to resist ischemia and oxygen-induced cardiomyocyte apoptosis by stabilizing mitochondria [48]. Studies have shown that enhanced mitochondrial function is essential for the differentiation of dental papilla cells into odontoblasts [82]. Thus, HSP10 may function together with HSP60 to stabilize mitochondria in tooth development. Similar to HSP25, HSPB5, HSPB6, and HSPB8 are able to interact with the cytoskeleton and prevent the entanglement and aggregation of intermediate filaments. However, HSPB6 and HSPB8 are less effective than HSP25 or HSPB5 in the protection of intermediate filaments [83]. It is speculated that HSPB5, HSPB6, and HSPB8 may play similar roles to HSP25 in tooth development but are less effective. The chaperone activity of HSP70 is inextricably linked to the J-domain in the HSP40 molecule, which pre-selects substrates and transfers them to HSP70, and stimulates HSP70 to hydrolyze ATP to form stable substrate binding. HSP40, therefore also plays an important role in physiological and pathological processes. In addition to being involved in the process by which HSP70 exerts its chaperone activity, HSP40 is also closely associated with the amyloid fibril depolymerization [84]. The HSP90 family also possesses many members, among which constitutively expressed HSP90, TRAP1 (tumor necrosis factor receptor-associated protein 1) in mitochondria, as well as Grp94/Gp96 in the ER are the focus of research. HSP90 plays a chaperone role with the help of ATP binding to different substrates. Due to the importance of HSP90 in maintaining protein activity, HSP90 is essential for the growth of cancer cells. In addition, studies have shown that HSP90 is associated with a variety of neurological and psychiatric diseases [85]. In inflammatory environments, IL-1β and TNF-α induce high levels of HSP90 expression. HSP90 interacts with actin depolymerization factor to activate cytoskeleton remodeling and enhance the differentiation ability of dental pulp stem cells and gingival mesenchymal stem cells [75]. HSP110, similar to other HSP, has the effect of preventing the accumulation of misfolded proteins, inhibiting apoptosis, and promoting cell proliferation through the Wnt and transcription factor STAT3 pathways [86]. HSP110 associates with axin in the Wnt pathway and regulates the degradation of the β-catenin degradation complex through protein phosphatase 2A (PP2A). When HSP110 is absent, β-catenin accumulation and target gene transcription are inhibited after Wnt stimulation, so HSP110 is essential in the Wnt/β-catenin pathway [87]. Because the Wnt/β-catenin pathway is an important pathway in tooth development, it can be speculated that HSP110 also plays an important role in tooth development.

## 7. Expression Patterns of HSP25, HSP60 and HSP70 in Tooth Development and Injury Repair

HSP25 is abundantly expressed as odontoblasts and ameloblasts mature and begin to synthesize secreted proteins in large quantities to reduce protein misfolding and accumulation. At the same time, because the tooth development process is accompanied by the movement of odontoblasts and ameloblasts, as well as elevated cell density, HSP25 may improve cellular tolerance to mechanical stimuli through p38 MAPK by assisting action. However, when ameloblasts and odontoblasts are not yet mature, HSP25 is not expressed, and the expression of HSP25 can be used as a marker for the conversion of these two cells from differentiation to maturation. During ameloblast differentiation, HSP25 is involved in the ERK/MAPK pathway to regulate ameloblast activity. HSP60 is expressed at the beginning of tooth development. Since HSP60 is mainly located in mitochondria, its potential role may be related to the maintenance of normal physiological functions of cells. In addition, HSP60 may regulate the proliferation and apoptosis of odontogenic cells through TLR4 and NF-κB pathways and affect the tooth development morphology. HSP70 is most widely expressed in tooth germs, especially in ameloblasts, where it plays a role in preventing protein misfolding, assisting protein transport, and maintaining cell homeostasis. The expression of GRP78 is also mainly located in cells with vigorous protein synthesis and secretion, and GRP78 may interact with DMP1 to promote the maturation and secretion of odontoblasts. Interestingly, there is also a distribution of GRP78 in the dentin matrix and alveolar bone matrix, and it is speculated that GRP78 is also involved in the mineralization process of the matrix in tooth development. The expression of HSC70 in tooth development may be related to the involvement of protein folding and degradation. In addition, HSC70 may assist protein transport from the cytoplasm to organelles or nuclei. However, there are few studies on HSC70 in tooth development, and the role of HSC70 in tooth development still needs further study to reveal its potential role. Why HSPs are differently expressed at different stages of tooth development remains to be answered. At different stages of tooth development, cells in the tooth germ undergo changes in morphology, function, as well as the surrounding environment, and the proteins expressed by the cells also vary. Different HSPs may therefore play a corresponding role when different proteins are expressed to assist teeth in completing final development.

However, in the injury repair of teeth, the expression of HSP25 is similar to the process of tooth development and begins to be abundantly expressed after the maturation of newly differentiated odontoblasts. Both HSP70 and GRP78 expression increased after a tooth injury, and the local inflammatory response after a tooth injury, as well as the enhancement of cellular activity during tooth repair, may be responsible for the increased expression of HSP70 and GRP78. However, little is known about the role of HSPs in the repair of dental injuries. Especially in the process of odontoblast differentiation and maturation, what role HSPs play and whether the role of HSPs during tooth injury repair is the same as its role during tooth development remains to be solved.

## 8. Conclusions

In conclusion, HSPs are involved in many processes in tooth development and injury repair, and although their specific mechanisms remain unclear, various types of HSPs have been reported to play a role in different cellular processes of odontoblasts and ameloblasts while maintaining the stability of related cells during tooth development and creating appropriate conditions for tooth development and repair. Future studies should aim to focus on exploring the mechanism of action and the combined effects with other factors to elucidate the potential of HSPs in tooth development-related diseases, tooth repair and odontogenic stem cells.

## Figures and Tables

**Figure 1 ijms-24-07455-f001:**
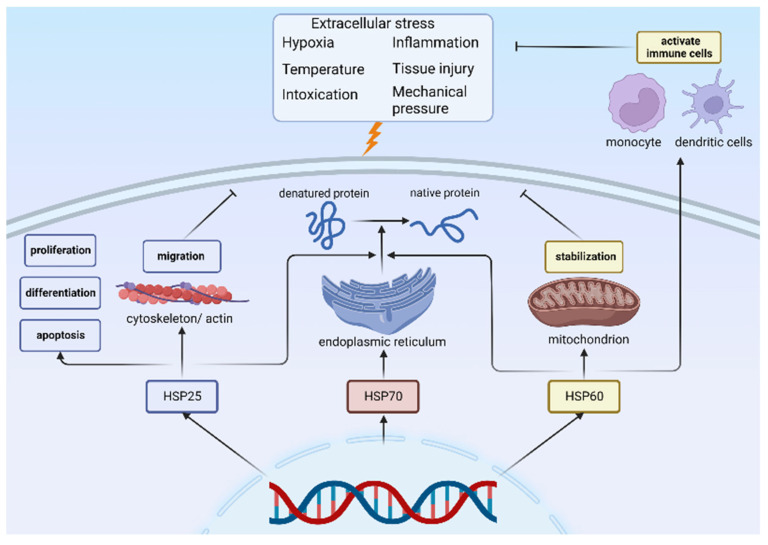
HSPs help cells resist external stimuli and maintain cell stability in different ways when cells encounter external stimuli. In addition, HSPs are also an important member of maintaining normal cell activity in healthy conditions. (┴): inhibit. Figures created with BioRender.com, accessed on 19 September 2022.

**Figure 2 ijms-24-07455-f002:**
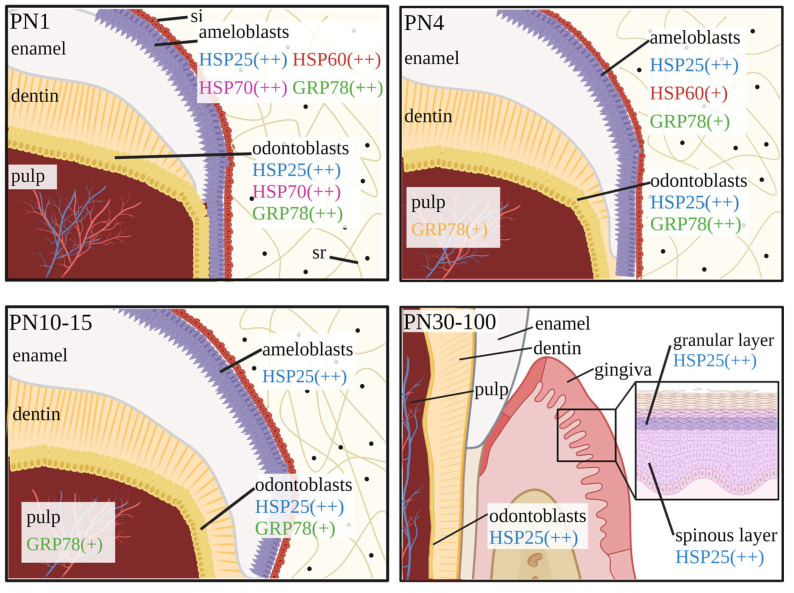
HSPs from postnatal day 1 to postnatal day 100 of mouse molar. The expression of HSP25 begins after odontoblasts and ameloblasts complete their differentiation, and HSP25 is strongly expressed when odontoblasts and ameloblasts secrete vigorously. When tooth development is basically completed, HSP25 is mainly expressed by odontoblasts. The expression of HSP60 and HSP70 peaked on PN2 and rapidly decreased. At the same time, the expression of GRP78 in ameloblasts is similar to that of HSP60. From PN4 to PN15, the expression of GRP78 continues in pulp, dentin and odontoblasts. At the time of tooth eruption, HSP25 is mainly localized in the spinous and granular layers of the gingival epithelium. (+): mild; (++): strong. Figures created with BioRender.com, accessed on 23 May 2022.

**Figure 3 ijms-24-07455-f003:**
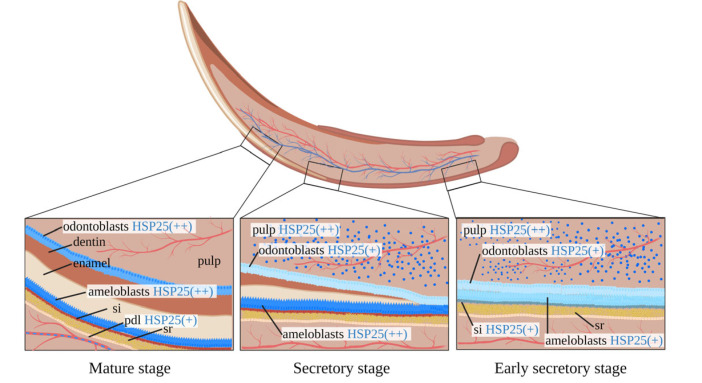
HSP25 in PN30 mouse incisors. From the early secretory stage to the mature stage, odontoblasts and ameloblasts gradually matured. Additionally, the secretory activity increased. Thus, the expression of HSP25 increased gradually. However, the expression of HSP25 in dental pulp decreased gradually. Additionally, the stratum intermedium shows a mild expression of HSP25 during the early secretory stage. Blood vessels in the periodontal ligament also express HSP25 mildly during the mature stage. pdl: periodontal ligament; si: stratum intermedium; sr: stellate reticulum. (+): mild; (++): strong. Figures created with BioRender.com, accessed on 23 May 2022.

**Figure 4 ijms-24-07455-f004:**
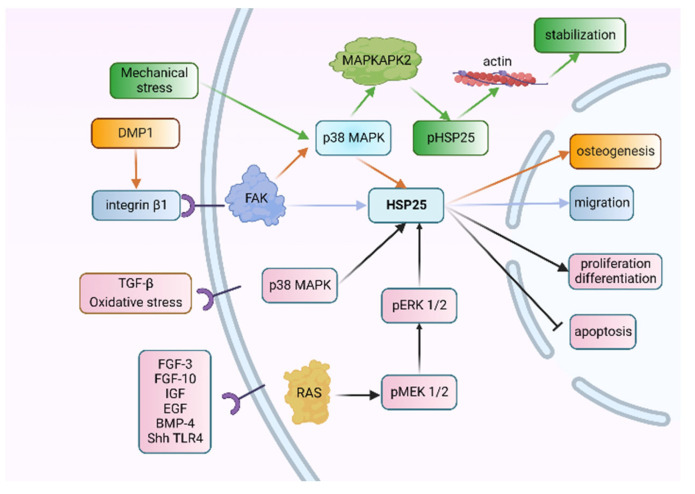
Signal pathways that HSP25 may participate in tooth development and damage repair. HSP25 is mainly involved in ERK/MAPK pathway and p38 MAPK pathway and participates in different cellular processes according to the received signals. Mechanical stress can phosphorylate HSP25 (pHSP25) through the p38 MAPK pathway, and pHSP25 binds to actin to stabilize cell structure. DMP1 upregulates HSP25 expression and promotes osteogenesis through the p38MAPK pathway. Integrin β1 can promote cell migration through HSP25. HSP25 receives signals from FGF, IGF, EGF, BMP-4, Shh, TLR4, TGF-β and oxidation stress through ERK/MAPK pathway and p38 MAPK pathway, thus promoting cell proliferation and differentiation and inhibiting apoptosis. Figures created with BioRender.com, accessed on 21 August 2022.

**Figure 5 ijms-24-07455-f005:**
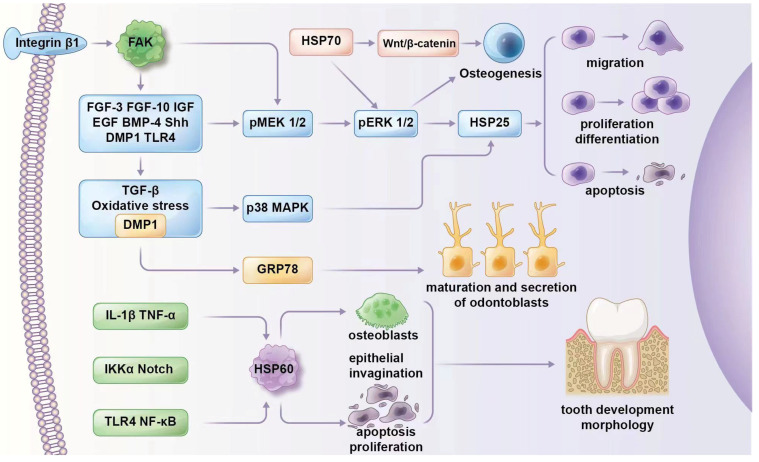
Signal pathways that HSPs may participate in tooth development and damage repair. HSP25 can promote cell migration, proliferation, and differentiation and inhibit apoptosis in the MAPK pathway. HSP70 and ERK participate in the process of osteogenesis. GRP78 and DMP1 promote the maturation and secretion of odontoblasts. HSP60 may affect tooth development and morphology by affecting epithelial invagination, as well as the proliferation and apoptosis of osteoclasts and tooth germ cells.

**Table 1 ijms-24-07455-t001:** Expression of heat shock protein 25 (HSP25) in the tooth germs of mandibular first molar in mice.

HSP25	Components of the Molar Tooth Germ								
	Epithelium						Interstitial Tissue		
Stage	iee	oee	sr	si	pab	ab	mc	pob	ob
Bud stage	/	/	/	/	/	/	/	/	/
Cap stage	-	-	-	/	-	-	/	/	/
Early bell stage	-	-	-	-	-	-	-	-	-
Late bell stage	-	-	+	-	+	+	+	+	+

Note: iee: inner enamel epithelium; oee: outer enamel epithelium; sr: stellate reticulum; si: stratum intermedium; pab: pre-ameloblasts; ab: ameloblasts; mc: mesenchymal cells; pob: pre-odontoblasts; ob: odontoblasts; (-): absent; (+): positive; (/): absent structure.

**Table 2 ijms-24-07455-t002:** Expression of heat shock protein 25 (HSP25) in the mandibular first molar on postnatal (PN) day 1–100 in rats.

HSP25	Components of the Molar Tooth Germ							
Stage (Day)	eo	dp	mc	pab	ab	pob	ob	oe
PN1	-	ph (++)	dp (-)	+	+	+	++	
			sob (+)					
PN5			ph (+)		++		++	
PN10			dp (+)		++		++	
PN15		ph (+)			++		++	+
		tr (+)						
PN30		ra (+)					tc (++)	
		tc (+)					tr (+/-)	
							pf (+/-)	
PN60–100							tc (++)	
							rt (++)	
							pf (++)	

Note: ph: pulp horn; dp: dental pulp; sob: sublayer of odontoblasts; tr: tooth root; ra: root apex; tc: tooth crown; pf: pulp floor; eo: enamel organ; mc: mesenchymal cells; pab: pre-ameloblasts; ab: ameloblasts; pob: pre-odontoblasts; ob: odontoblasts; oe: oral epithelium; (-): absent; (+): mild; (++): strong.

**Table 3 ijms-24-07455-t003:** Expression of heat shock protein 25 (HSP25) in the incisors of postnatal (PN) day 30 rats.

HSP25	Components of the Incisors				
Stage	pdl	dp	si	ab	ob
Early secretory stage	-	++	+	+	+
Secretory stage	-	++	-	++	
Mature stage	blood vessels (+)	-		ra (++) sa (+)	++

Note: pdl: periodontal ligament; dp: dental pulp; si: stratum intermedium; ab: ameloblasts; ob: odontoblasts; (-): absent; (+): mild; (++): strong.

**Table 4 ijms-24-07455-t004:** Expression of heat shock protein 60 (HSP60) in mouse mandibular incisor tooth germs.

HSP60	Components of the Incisors Germ											
	Epithelium								Interstitial Tissue			
Stage	oe	eb	iee	oee	ek	si	pab	ab	dp	df	pob	ob
Initial Stage	+	+										
Bud Stage			++	++	++				+	+		
Cap stage			++	++	++				+	+		
Bell Stage			++	++		++	++	++		+	+	+

Note: oe: oral epithelium; eb: epithelium band; iee: inner enamel epithelium; oee: outer enamel epithelium; ek: enamel knot; si: stratum intermedium; pab: pre-ameloblasts; ab: ameloblasts; dp: dental pulp; df: dental follicle; pob: pre-odontoblasts; ob: odontoblasts; (+): mild; (++): strong.

**Table 5 ijms-24-07455-t005:** Expression of heat shock protein 70 (HSP70) in the epithelial and mesenchymal parts of the tooth germ between the 9th and 20th developmental week [69].

HSP70	Tooth Germ Parts											
	Epithelial									Epithelial		
Age (Weeks)	tb	dl	iee	oee	ek	cl	sr	si	pa	dp	df	pob
9	+++	+++	+++	+++	+	/	/	/	/	-	-	/
12	/	+++	+++	+++	+++	++	++	/	/	-	-	/
14	/	++	++/+++	+++	/	++	+	+	++	+	-	/
20	/	+	++	+	/	+++	++	++	+++	+	-	++

Note: tb: tooth bud; dl: dental lamina; iee: inner enamel epithelium; oee: outer enamel epithelium; ek: primary enamel knot; cl: cervical loop; sr: enamel reticulum; si: stratum intermedium; pa: pre-ameloblasts; dp: dental papilla; df: dental follicle; pob: pre-odontoblasts; (-): absent; (+): mild; (++): moderate; (+++): strong; (/): absent structure.

## Data Availability

Not applicable.

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
