# Peer review of "Heat Shock Proteins in Tooth Development and Injury Repair"

_ijms, 2023, doi:10.3390/ijms24087455_

Round 1
Reviewer 1 Report
In their paper, the authors have provided a comprehensive review of heat shock protein expression and their role in tooth development and injury repair.
There are no major issues in the provided review, while the quality of the publication can be improved by comprehension of the provided text and shortening the length of the paper. Minor spell check and grammar corrections must be made in a few instances.
Author Response
Comments: In their paper, the authors have provided a comprehensive review of heat shock protein expression and their role in tooth development and injury repair. There are no major issues in the provided review, while the quality of the publication can be improved by comprehension of the provided text and shortening the length of the paper. Minor spell check and grammar corrections must be made in a few instances.
Author’s Response: Thank you for taking the time to review our paper on the role of heat shock proteins in tooth development and injury repair. We are pleased to hear that you found our review to be comprehensive and without major issues. We appreciate your feedback regarding the quality of the publication and we make sure to conduct a thorough spell check and correct any grammatical errors.

Reviewer 2 Report
Review note
This paper reviews an important topic – the role of Heat shock proteins in tooth development and injury repair. The authors reviewed a nice set of literatures to address the topic, which will help further translational and clinical application. The structure is generally clear and logically organized. However, to grow into a publication, I think there are some issues the authors need to address.
1. The authors highlighted HSP25,60, and 70 in this review, thus, these proteins should be emphasized in the introduction.
2. The similarity and the difference between tooth development and injury repair should also be involved in the introduction, since the authors discussed these two process in the same paper.
3. Section 7 is also about HSP25,60, and 70, thus, I recommend the authors to put it after the section for HSP 70, or integrate them in each section respectively.
4. Is there any literatures on the roles of HSP10, HSPB5, HSPB6, HSPB8, HSP90 and HSP110 in injury repair?
In general, this review is clear-cut and reflects an important issue. I hope the author(s) could find some of the above discussions helpful for improving the paper.
Author Response
Comments: This paper reviews an important topic – the role of Heat shock proteins in tooth development and injury repair. The authors reviewed a nice set of literatures to address the topic, which will help further translational and clinical application. The structure is generally clear and logically organized. However, to grow into a publication, I think there are some issues the authors need to address.
- The authors highlighted HSP25,60, and 70 in this review, thus, these proteins should be emphasized in the introduction.
Author’s Response: We thank the reviewer for your suggestion. We have added a sentence to emphasize HSP25,60 and 70 in the introduction. Relevant contents are highlighted in YELLOW (Please see Introduction Page 2 Paragraph 1 line 58-60).
- The similarity and the difference between tooth development and injury repair should also be involved in the introduction, since the authors discussed these two processes in the same paper.
Author’s Response: Thanks to your constructive advice. We have added relevant contents in the introduction. Also, the figure in Page 1 shows the similarity and the difference between tooth development and injury repair. Relevant contents are highlighted in YELLOW (Please see Introduction Page 2 Paragraph 1 line 55-57).
- Section 7 is also about HSP25,60, and 70, thus, I recommend the authors to put it after the section for HSP 70, or integrate them in each section respectively.
Author’s Response: We appreciate the reviewer’s suggestion. We have carefully re-considered the arrangement of the article structure and tried to integrate Section 7 in each section. However, we found that the Expression pattern of HSPs cannot be clearly displayed in this constructure. Therefore, at the end of the review, we reviewed the expression patterns of HSP25,60,70 as a whole to make the article more complete. However, we still appreciate the excellent suggestions you have provided, which will benefit us greatly in our future writing.
- Is there any literatures on the roles of HSP10, HSPB5, HSPB6, HSPB8, HSP90 and HSP110 in injury repair?
Author’s Response: We appreciate the reviewer for your effort to reviewing our manuscript. Currently, the roles of HSPs in tooth injury repair has not been fully studied, so there are no relevant literatures on the roles of above-mentioned HSPs in injury repair.
In general, this review is clear-cut and reflects an important issue. I hope the author(s) could find some of the above discussions helpful for improving the paper.

Reviewer 3 Report
Dear authors, Interesting paper and topic. It covers well the HSPs and their putative role in tooth development and injury repair. I have not too much to day, but I would strongly suggest the following points which I really think can strenghtenen the article:
1- Improve the abstract
2- add a section about HSP, Folding, Drugs, and Nanotechnology: Possible Mehods and Applications in Dentistry
> cite the following key references:
A/ Menaa B, Miyagawa Y, Takahashi M, et al. Bioencapsulation of apomyoglobin in nanoporous organosilica sol-gel glasses: influence of the siloxane network on the conformation and stability of a model protein. Biopolymers. 2009 Nov;91(11):895-906. doi: 10.1002/bip.21274.
B/ Arif W, Rana NF, Saleem I, et al. Nadeem AY. Antibacterial Activity of Dental Composite with Ciprofloxacin Loaded Silver Nanoparticles. Molecules. 2022 Oct 24;27(21):7182. doi: 10.3390/molecules27217182. .
C/ Saleem I, Rana NF, Tanweer T, et al. Effectiveness of Se/ZnO NPs in Enhancing the Antibacterial Activity of Resin-Based Dental Composites. Materials (Basel). 2022 Nov 6;15(21):7827. doi: 10.3390/ma15217827.
D/ Azhar S, Rana NF, Kashif AS, et al. DEAE-Dextran Coated AgNPs: A Highly Blendable Nanofiller Enhances Compressive Strength of Dental Resin Composites. Polymers (Basel). 2022 Aug 2;14(15):3143. doi: 10.3390/polym14153143.
E/ Tanweer T, Rana NF, Saleem I, et al. Dental Composites with Magnesium Doped Zinc Oxide Nanoparticles Prevent Secondary Caries in the Alloxan-Induced Diabetic Model. Int J Mol Sci. 2022 Dec 14;23(24):15926. doi: 10.3390/ijms232415926.
F/ Sasaki Y, Akiyoshi K. Nanogel engineering for new nanobiomaterials: from chaperoning engineering to biomedical applications. Chem Rec. 2010 Dec;10(6):366-76. doi: 10.1002/tcr.201000008.
G/ Morimoto N, Endo T, Iwasaki Y, Akiyoshi K. Design of hybrid hydrogels with self-assembled nanogels as cross-linkers: interaction with proteins and chaperone-like activity. Biomacromolecules. 2005 Jul-Aug;6(4):1829-34.
BEST,
The reviewer
Author Response
Comments: Dear authors, interesting paper and topic. It covers well the HSPs and their putative role in tooth development and injury repair. I have not too much to day, but I would strongly suggest the following points which I really think can strengthen the article:
1- Improve the abstract.
Author’s Response: Thanks for your kindly proposal. We have checked and revised the abstract properly.
2- add a section about HSP, Folding, Drugs, and Nanotechnology: Possible Methods and Applications in Dentistry.
> cite the following key references:
A/ Menaa B, Miyagawa Y, Takahashi M, et al. Bioencapsulation of apomyoglobin in nanoporous organosilica sol-gel glasses: influence of the siloxane network on the conformation and stability of a model protein. Biopolymers. 2009 Nov;91(11):895-906. doi: 10.1002/bip.21274
B/ Arif W, Rana NF, Saleem I, et al. Nadeem AY. Antibacterial Activity of Dental Composite with Ciprofloxacin Loaded Silver Nanoparticles. Molecules. 2022 Oct 24;27(21):7182. doi: 10.3390/molecules27217182
C/ Saleem I, Rana NF, Tanweer T, et al. Effectiveness of Se/ZnO NPs in Enhancing the Antibacterial Activity of Resin-Based Dental Composites. Materials (Basel). 2022 Nov 6;15(21):7827. doi: 10.3390/ma15217827
D/ Azhar S, Rana NF, Kashif AS, et al. DEAE-Dextran Coated AgNPs: A Highly Blendable Nanofiller Enhances Compressive Strength of Dental Resin Composites. Polymers (Basel). 2022 Aug 2;14(15):3143. doi: 10.3390/polym14153143
E/ Tanweer T, Rana NF, Saleem I, et al. Dental Composites with Magnesium Doped Zinc Oxide Nanoparticles Prevent Secondary Caries in the Alloxan-Induced Diabetic Model. Int J Mol Sci. 2022 Dec 14;23(24):15926. doi: 10.3390/ijms232415926
F/ Sasaki Y, Akiyoshi K. Nanogel engineering for new nanobiomaterials: from chaperoning engineering to biomedical applications. Chem Rec. 2010 Dec;10(6):366-76. doi: 10.1002/tcr.201000008.
G/ Morimoto N, Endo T, Iwasaki Y, Akiyoshi K. Design of hybrid hydrogels with self-assembled nanogels as cross-linkers: interaction with proteins and chaperone-like activity. Biomacromolecules. 2005 Jul-Aug;6(4):1829-34.
Author’s Response: Thank you for your valuable suggestion. We have added the information on HSP's possible methods and applications in dentistry. Relevant contents are highlighted in GREEN (Please see section 5 “Research prospect of HSPs in tooth development and injury repair” Page 13).

Reviewer 4 Report
Dear Authors
The topic that You disscusse is of very high importance. Only some small issues should be concerned. Maybe You could consider some change in the title to make it more eye-catching.
The results are nicely presented and comprehe sible.
The choice of references is correct and well presented in the context of Your research.
I hope You will find these observations useful.
Best regards.
Author Response
Comments: The topic that you discuss is of very high importance. Only some small issues should be concerned. Maybe You could consider some change in the title to make it more eye-catching. The results are nicely presented and comprehensible. The choice of references is correct and well presented in the context of Your research.
Author’s Response: Thank you for your feedback on our paper. We are glad to hear your positive comment on our work. Regarding the suggestion to make changes to the title, we have made several attempts, but did not achieve better results. Therefore, the original title was kept. However, we still appreciate the excellent suggestions you have provided, which will benefit us greatly in our future writing.
